# Outcomes Addressed by Whole-Body Electromyostimulation Trials in Sportspeople and Athletes—An Evidence Map Summarizing and Categorizing Current Findings

**DOI:** 10.3390/sports13090302

**Published:** 2025-09-02

**Authors:** Svenja Reinhardt, Joshua Berger, Matthias Kohl, Simon von Stengel, Michael Uder, Wolfgang Kemmler

**Affiliations:** 1Institute of Radiology, University Hospital Erlangen, Henkestrasse 91, 91052 Erlangen, Germany; svenja.reinhardt@fau.de (S.R.);; 2German Research Center for Artificial Intelligence (DFKI), 66113 Saarbrücken, Germany; 3Department of Medical and Life Sciences, University of Furtwangen, 78056 Schwenningen, Germany; matthias.kohl@hs-furtwangen.de; 4Institute of Medical Physics, Friedrich-Alexander University of Erlangen-Nürnberg, 91052 Erlangen, Germany

**Keywords:** whole-body electrostimulation, electromyostimulation, sportspeople, athletes, performance, regeneration, safety

## Abstract

Whole-body electromyostimulation (WB-EMS) is a time-efficient, joint-friendly, and highly customizable training technology that particularly attracts sportspeople and athletes looking to enhance performance, accelerate regeneration, and prevent injuries with WB-EMS. Based on a systematic review of the literature, the present evidence map aimed to provide an overview of outcomes addressed by WB-EMS in exercising cohorts of different levels. In summary, the search identified 34 research projects with 39 studies and 43 publications that addressed 79 outcome categories (e.g., isometric strength) with more than 300 single outcomes (e.g., isometric strength of leg extensors). Thirty-one studies focused on performance-related outcomes, four studies addressed regeneration-related outcomes, and eight studies reported outcomes related to anthropometry. A further 14 studies reported health- and safety-related outcomes. Twenty-five of the 31 studies that reported performance parameters addressed strength, ten power, 18 jumping, ten sprinting, six agility, six endurance, five anaerobic power, and one each flexibility or balance, and five studies reported sport-specific performance outcomes (e.g., shot velocity). Apart from outcomes concerning injury prevention or sport-specific complaints, there are in particular evidence gaps relating to the acute effects of WB-EMS on regeneration, particularly with respect to muscle recovery. Semiprofessionals/professionals were rarely addressed, and if so, primarily cohorts from team sports were evaluated, while no study focused on elite strength, endurance, or precision sports athletes.

## 1. Introduction

Due to its ability to stimulate all major muscle groups simultaneously, albeit with dedicated intensity and without high mechanical strain, whole-body electromyostimulation (WB-EMS) can be considered a time-effective, joint-friendly, and highly customizable exercise training technology [1,2]. This by and large refers to the standard WB-EMS protocol [3,4,5], predominantly applied in the fitness and health domain, usually without intense superimposed (voluntary) exercises and conducted 1–2 × 20 min/week in an individualized and closely supervised setting. Nevertheless, the general efficiency of WB-EMS may also attract sportspeople at different levels to use WB-EMS to improve their performance, prevent injury, or speed up recovery. However, summarizing the present research in this area is problematic, particularly due to the heterogeneity of the studies with respect to study design, performance level/athletic background, type of exercise/discipline, WB-EMS approach, control groups, and endpoint measures. An evidence map based on a systematic and comprehensive review of the WB-EMS literature might be helpful to structure the available evidence on WB-EMS in sportspeople and athletes and to identify gaps in knowledge and/or areas for future research. Briefly, evidence mapping is a procedure for evidence synthesis that summarizes and organizes research findings on a specific topic, particularly in order to show the amount and characteristics of studies in the given research topic using a comprehensive and clearly arranged user-friendly format [6]. Compared to the more specific eligibility criteria of systematic reviews and meta-analyses, evidence maps cover a broader range of study designs, populations, interventions, and outcomes. Further, unlike other forms of evidence synthesis, the purpose of evidence mapping is not to evaluate the effectiveness of interventions [7] but to present a comprehensive overview of all the available studies in the field. In a recent evidence map on WB-EMS outcomes, we focused on largely sport-inactive cohorts [3,5] with predominately isolated WB-EMS application, i.e., the main area of commercial WB-EMS application [8]. In that project, however, we decided not to include sportspeople/athletes for several reasons. Apart from more specific performance outcomes and possibly diverging physiologic responses to WB-EMS related to the enhanced training status, the most striking reason was that the (superimposed) WB-EMS approach for sportspeople/athletes differs significantly from protocols usually applied in the health and fitness domain of non-athletes.

In the present evidence map on study outcomes addressed by WB-EMS in sportspeople and athletes, our aim is to provide a comprehensive overview, not least in order to identify gaps in the literature and prevent duplications of studies that might address the same outcome in similar cohorts.

## 2. Materials and Methods

The literature search for this systematic review and evidence map followed the Preferred Reporting Items for Systematic Reviews and Meta-Analyses (PRISMA) Statement. The completed PRISMA checklist is provided in the Appendix A. The study was registered on 2 February 2025 under the PROSPERO ID CRD420250646327.

### 2.1. Eligibility Criteria

#### 2.1.1. Population

Cohorts comprising athletes [9], advanced sportspeople, or physical education/sport students were included. Recreational/hobby sportspeople were accepted when exercising for >2 years with a weekly training frequency of two or more sessions per week.

#### 2.1.2. Intervention

Only studies that applied whole-body electromyostimulation, defined as “simultaneous application of electric stimuli via at least six current channels or participation of all major muscle groups, with a current impulse effective to trigger muscular adaptations” [1,4] were included. On the other hand, studies that applied local EMS or focused on single muscle groups or regions (e.g., thighs) were not considered. The decision to exclude local EMS applications was based on the aspect that we assumed that a technology that stimulates all major muscle groups simultaneously, but with dedicated intensity, has a more comprehensive range of applications.

#### 2.1.3. Comparators

The presence of a control group was not considered as an eligibility criterion.

#### 2.1.4. Outcomes

All types of outcomes were accepted.

#### 2.1.5. Study Design

All study designs except single case studies, review articles, editorials, conference abstracts, and letters were included. In parallel, bachelor or master theses were excluded, while dissertations were included where there were no corresponding publications.

### 2.2. Information Sources

Study reports from the five electronic databases, the Cumulative Index to Nursing & Allied Health [CINAHL via Ebsco Host], the Cochrane Central Register of Controlled Trials [CENTRAL], Medline [PubMed], SPORTDiscus (via Ebsco Host), Web of Science (via Clarivate) published from their initiation up to 6 March 2025 were searched without language restrictions (Figure 1).

### 2.3. Literature Search

A standard protocol for this search was developed, and a controlled vocabulary (MeSH term for MEDLINE, CINAHL^®^ Subject Headings for CINAHL) was applied. Keywords and their synonyms were used by applying the following queries: WB-EMS OR “whole body electro myostimulation” OR electromyostimulation OR “electrical muscle stimulation” OR electro-myo-stimulation OR electrostimulation OR “integral electrical stimulation” OR “whole-body electrical muscle stimulation” AND athletic OR athlete OR sport OR performance OR trained. To identify all relevant studies, reference lists of eligible articles were also screened.

### 2.4. Selection Process

Two reviewers (SB, WK) independently screened titles, abstracts, and full texts against the eligibility criteria listed below. Disagreements were resolved by discussion or by including a third reviewer (SvS) and a majority decision. In the case of missing, incomplete, or unclear data, the authors were contacted a maximum of three times.

### 2.5. Data Management

Search results were downloaded, and title and abstract screening as well as full-text screening were conducted with Endnote. Duplicates were identified and excluded using the method suggested by Bramer et al. [10]. In cases of multiple publications of the same project that addressed identical cohorts and outcomes, only the main publication was included.

### 2.6. Data Extraction

Two authors separately and independently extracted data from the included studies using a Microsoft Excel table in the approach listed above. The sheets of the table were separated into 4 categories: (a) study and publication characteristics, with, for example, first author, year, country of the publication, study design, study type, number of study arms, sample size, comparator, and methodologic quality of the studies if applicable; (b) cohort/participant characteristics, with, for example, gender, age, body height, body mass, body mass index, exercise status, and type of exercise/discipline. Of importance, exercise status was categorized into hobby sportspeople (≥2 years, 2 sessions/week without competitions), advanced sportspeople (sportspeople who participate in competitions and/or regular league games and physical education/sport students), and semi-professional/full-professional “athletes”; (c) intervention characteristics, e.g., study/intervention length, mixed or isolated WB-EMS protocols, superimposed exercise, exercise volume (weekly training frequency, duration of the session), details of the WB-EMS protocol (impulse type, frequency, width, length, break, and intensity); and (d) loss to follow-up, attendance, and adverse effects.

### 2.7. Quality Assessment

Two independent reviewers assessed eligible studies (i.e., RCT, RCOT) for risk of bias using the Physiotherapy Evidence Database (PEDro) Scale Risk of Bias Tool [11] that specifically covers exercise studies. In case of inconsistencies (n = 7), a third independent reviewer made the decision. Following Ribeiro de Avila [12] we classified the methodological quality of the studies as follows: <5 score points: low, 5–7 score points: moderate, and >7 score points: high. Of note, we also rated non-randomized controlled trials (n = 2, Table 1) and randomized controlled cross-over studies, although the PEDro score is not ideally suited for these types of studies.

### 2.8. Data Synthesis

Study, publication, cohort, participant, and intervention characteristics are displayed in tables. Bubble charts with 4 dimensions were created for cohorts and outcomes. The *y*-axis consistently presents the number of studies the *x*-axis focuses on either cohorts included in the projects (Figure 2), study outcomes (Figure 3), or, more detailed, performance-related outcomes (Figure 4). The size of the bubbles indicates the athletic status of the cohorts included in the projects (i.e., hobby sportspeople vs. advanced sportspeople vs. semiprofessionals/professionals); the color of the study represents whether the project focuses on acute or longitudinal effects. To provide a quick overview, the results were divided into the categories “Anthropometry”, “Performance”, “Regeneration”, “Health/Safety”, and “Explanations”. Outcomes related to “performance” were categorized as “strength-related outcomes”, “power-related” outcomes”, “jumping-related outcomes”, “sprinting-related outcomes”, “agility-related outcomes”, “endurance-related outcomes”, “anaerobic power-related outcomes”, flexibility-related outcomes”, “balance-related outcomes”, and outcomes related to sport-specific performance.

**Figure 1 sports-13-00302-f001:**
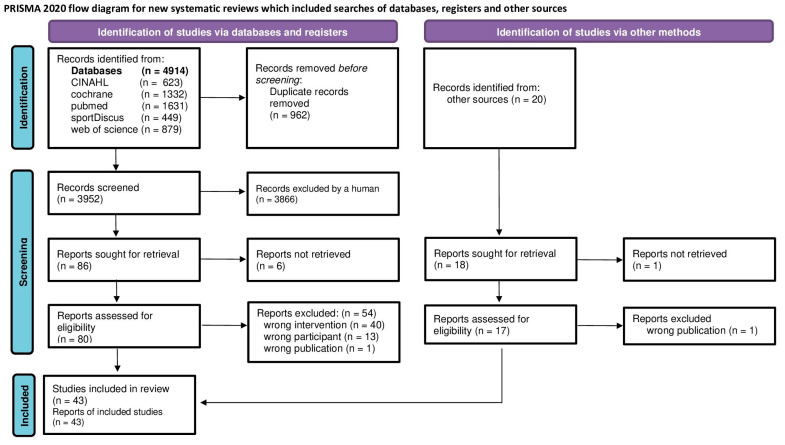
Flow diagram of search process according to PRISMA [13].

## 3. Results

Of the 4914 records, 34 research projects with 39 studies and 43 publications [14,15,16,17,18,19,20,21,22,23,24,25,26,27,28,29,30,31,32,33,34,35,36,37,38,39,40,41,42,43,44,45,46,47,48,49,50,51,52,53,54,55,56] are finally included in the present evidence map (Figure 1; Table A1). Studies reporting different outcomes of the same cohort in multiple publications are summarized in Table 1 and Table 2 and correspondingly considered in the analysis. Two other projects that separated outcomes into two publications each ([14,15] and [20,26]) reported data of diverging participants and are, thus, not summarized for tables or analysis. Lastly, one publication [37] reported results of two different interventions with the identical cohort on diverging outcomes. In summary, 38 projects are listed in Table 1 but 39 cases are included in the analysis.

### 3.1. Publication and Study Characteristics

Table 1 displays the publication and study characteristics of the included trials. Most projects focus on longitudinal changes in outcomes (n = 24) 15 studies addressed acute changes. Most longitudinal studies (n = 23) applied a randomized controlled design. In parallel, 12 studies that focus on acute effects used a randomized cross-over design, one study applied a randomized controlled design [18], one study applied a non-randomized controlled design [17], and one study [16] used a single-group design. The number of study groups per trial varied between one (e.g. [16]) and four [22], the total number of participants included in the trials ranged from nine [20] to 60 [31,32]. Most studies were conducted in Germany (n = 16) and Spain (n = 9).

**Table 1 sports-13-00302-t001:** Study and participant characteristics of the studies.

	Author	Publication Year	Study Design	Study Groups/Conditions [n]	Total Sample Size [n]	Gender	Age [Years]	BMI (kg/m^2^) ^1^	Status	Discipline	Methodological Quality (PEDro)
1	Amaro-Gahete et al. [14]	2018	RCT, longitudinal	2	14	m	27 ± 7	23 ± 3	Hobby sportspeople	Runners (vs. CG)	4
2	Amaro-Gahete et al. [15]	2018	RCT, longitudinal	2	14	m	27 ± 7	24 ± 3	Hobby sportspeople	Runners (vs. EMS)	4
3	Berger et al. [16]	2019	SGD, acute effects	1	52	m + w	24 ± 3	24 ± 2	Advanced sportspeople	Allrounders	na
4	Boccia et al. [17]	2017	NRCT, acute effects	2	10	m + w	24 ± 3	24	Hobby sportspeople	Allrounders	5
5	Buonseno et al. [18]	2023	RCT, acute effects	2	16	w	22 ± 2	ng	Advanced sportspeople	Allrounders	5
6	De Arrilucea et al. [20]	2025	RCOT, acute effects	2	10	m	20 ± 1	23 ± 3	Semi/full professionals	Soccer players	5
7	De La Camara et al. [21]	2018	RCOT, acute effects	3	9	m	21. ± 1	22 ± 3	Advanced sportspeople	Allrounders	7
8	Dörmann et al. [22]	2019	RCT, longitudinal	2	28	w	21 ± 2	22 ± 2	Advanced sportspeople	Allrounders	4
9	Dote-Montero et al. [23]	2021	RCOT, acute effects	4	10	m	23 ± 4	24 ± 3	Hobby sportspeople	Allrounders	5
10	D‘Ottavio et al. [19]	2019	RCT, longitudinal	3	22	m + w	26 ± 3	22 ± 3	Advanced sportspeople	Allrounders	4
11	Evangelista et al. [24]	2019	RCT, longitudinal	3	58	m + w	27 ± 4	25	Hobby sportspeople	Allrounders	5
12	Fernández-Elías et al. [26]	2022	RCOT, acute effects	3	20	m + w	ng	ng	Hobby sportspeople	Allrounders	4
13	Fernández-Elías et al. [25]	2024	RCOT, acute effects	2	12	m	22 ± 2	23 ± 3	Semi/full professionals	Soccer players	5
14	Filipovic et al. [29]	2016	RCT, longitudinal	2	22	m	26 ± 3	24 ± 2	Semi/full professionals	Soccer players	4
15	Filipovic et al. [27,28]	2019	RCT, longitudinal	3	28/30	m	23 ± 4	24 ± 2	Semi/full professionals	Soccer players	6
16	Hussain et al. [30]	2019	RCT, longitudinal	2	40	w	young	ng	Advanced sportspeople	Softball players	3
17	Hussain et al. [31,32]	2021, 2021	RCT, longitudinal	3	60	w	24 ± 2	22	Advanced sportspeople	Softball players	5
18	Ilbak et al. [33]	2022	RCT, longitudinal	2	20	m	15–20	22	Semi/full professionals	Basketball players	4
19	Jawad et al. [34]	2020	RCT, longitudinal	2	10	m	ng	ng	Semi/full professionals	Soccer players	3
20	Kacoglu et al. [35]	2021	RCT, longitudinal	2	38	m + w	22 ± 3	22 ± 2	Advanced sportspeople	Allrounders	4
21	Kemmler et al. [36]	2012	RCOT, acute effects	2	19	m	26 ± 5	24 ± 2	Advanced sportspeople	Allrounders	5
22	Kemmler et al. [37]	2019	RCOT, acute effects	3	19	m	29 ± 5	24 ± 2	Advanced sportspeople	Allrounders	8
23	Ludwig et al. [38]	2020	NRCT, longitudinal	2	30	m	15–17	22	Advanced sportspeople	Soccer players	4
24	Martín-Simón et al. [39]	2022	RCT, longitudinal	2	20	m + w	19–25	23	Advanced sportspeople	Allrounders	4
25	Mathes et al. [40]	2017	RCT, longitudinal	2	24	m	23 ± 5	23	Advanced sportspeople	Allrounders	5
26	Micke et al. [41]	2018	RCT, longitudinal	2	18	m	23 ± 3	22 ± 2	Advanced sportspeople	Allrounders	5
27	Qin et al. [42]	2022	RCT, longitudinal	2	20	m	25 ± 4	24 ± 1	Hobby sportspeople	Allrounders	6
28	Rappelt et al. [43]	2023	RCT, longitudinal	2	26	m + w	21 ± 2	22	Advanced sportspeople	Allrounders	4
29	Sadeghipour et al. [44]	2021	RCT, longitudinal	3	30	w	26 ± 2	23 ± 2	Hobby sportswoman	Allrounders	5
30	Schuhbeck et al. [45]	2019	RCT, longitudinal	2	30	m	28 ± 8	24	Advanced sportspeople	Ice-hockey players	5
31	Teschler et al. [46]	2018	RCOT, acute effects	1	16	m + w	34 ± 10	24 ± 2	Advanced sportspeople	Allrounders	6
32	Wahl et al. [49]	2012	RCOT, acute effects	1	10	m	25 ± 3	23	Advanced sportspeople	Allrounders	5
33	Wahl et al. [47,48]	2014/2015	RCOT, acute effects	1	13	m	25 ± 4	23	Advanced sportspeople	Allrounders	5
34	Wirtz et al. [51,52]	2015, 2016	RCT, longitudinal	2	20	m	22 ± 2	24	Advanced sportspeople	Allrounders	4
35	Wirtz et al. [50]	2020	RCT, longitudinal	3	28	m	23 ± 4	24	Advanced sportspeople	Soccer players	4
36	Zhang et al. [53]	2021	RCT, longitudinal	2	10	W	27 ± 4	22	Hobby sportspeople	Resistance exercise	4
37	Zink-Rückel et al. [54]	2019	RCOT, acute effects	2	20	m	37 ± 14	26 ± 5	Advanced sportspeople	Golf players	7
38	Zink-Rückel et al. [55,56]	2021, 2021	RCT, longitudinal	2	54	m	43 ± 14	27 ± 4	Hobby sportspeople	Golf players	7

NCT: Non-randomized controlled trial; RCT: randomized controlled trial; RCOT: randomized cross-over trial; SGD: single group design. ^1^ If not specified, BMI was calculated based on body height and body mass.

### 3.2. Participant Characteristics

Table 1 shows the participant characteristics of the included projects. Most importantly, 10 projects included hobby sportspeople [14,15,17,23,24,26,42,44,53,55,56], 23 projects focused on advanced sportspeople such as physical education/sport students (n = 16), and 6 studies included semi-professional/professional athletes [20,25,27,28,29,33,34]. Participants of twenty-three studies can be considered as allrounders (including physical education/sport students); two studies focused on runners [14,15], two projects on golf players [54,55,56], and one study involved persons engaged in resistance type exercise [53]. Eleven projects [20,25,27,28,29,30,31,32,34,38,45,50] focused on team sports (soccer: n = 7, softball: n = 2, basketball: n = 1, ice hockey: n = 1). Twenty-three projects included only males, six only females, and nine focused on mixed cohorts (Table 1). Participants were usually aged between 20 and 30 years old; only three projects [46,54,55,56] included participants in their fourth decade of life. All projects focused on healthy adults; cohorts comprising people with overweight (i.e., BMI > 25 kg/m^2^) were rare [54,55,56]. Figure 2 shows the participant characteristics of studies included in the present evidence map categorized (*x*-axis) according to type of exercise.

**Figure 2 sports-13-00302-f002:**
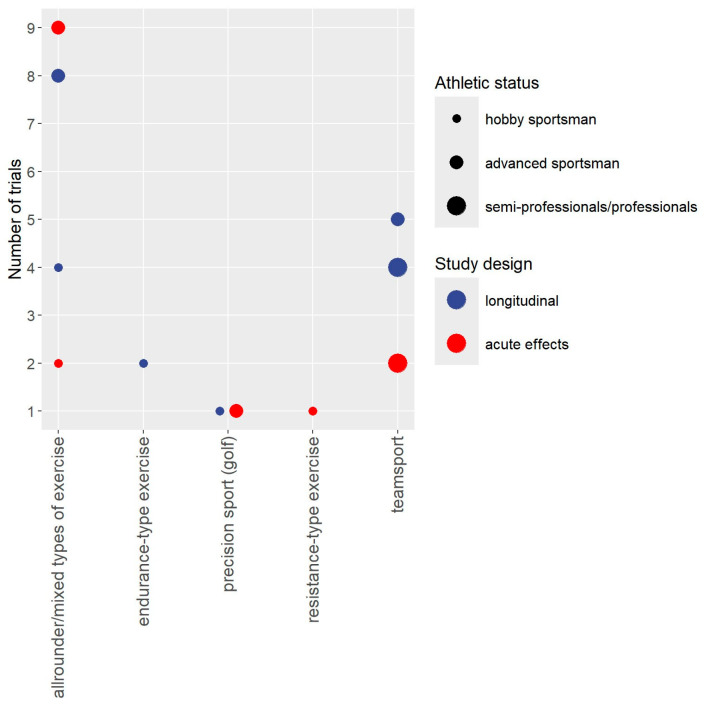
Participant characteristics of studies are included in the evidence map. Note: Physical education/sport students were considered “allrounders” and “advanced sportspeople”.

### 3.3. Exercise and WB-EMS Characteristics

Table 2 and Table 3 display exercise and WB-EMS characteristics of the studies. As an eligibility criterion, all studies applied WB-EMS. However, some projects focused on the stimulation of a limited number of areas, frequently the lower extremities (including gluteal muscles) [22,24,34,35,40,43,47,48,49,51,52] due to their intention to address predominately locally caused performance parameters. Few other projects focused on outcomes related to local muscle groups (e.g., leg strength or power), however, applied a WB-EMS approach (e.g. [18,23]). Although sometimes difficult to categorize [34], the majority of projects (n = 28) applied superimposed WB-EMS, i.e., voluntary exercise with moderate to high exercise intensity with added WB-EMS. Apart from a few studies [14,15,44], most longitudinal projects conducted WB-EMS programs without supplementary (i.e., additional) exercise protocols, i.e., combinations of WB-EMS sessions and sessions of other types of exercise. Study duration of the longitudinal studies ranged from four [22,40,43] to 16 weeks [55,56], and weekly exercise frequency was between one and 3.5 sessions [40]. The length of the WB-EMS application varied considerably between nine [27,28,29] and 60 min [40]; however; most projects applied sessions of 15–25 min. All projects used bipolar (biphasic) stimulation currents in the low-frequency range [4] (Table 2 and Table 3). Apart from three projects [14,15,21,37] that stimulated with impulse frequencies of 1–12 Hz and two studies that applied 100 [35] and 120 Hz [39], all the other studies used a standard WB-EMS protocol [37] with impulse frequencies of 80–90 Hz and an impulse breath of 300–400 µs. Only a few projects [21,23,24,26,37,47,48] applied continuous WB-EMS protocols (i.e., with WB-EMS application over the entire session), while most projects used programs with short bouts of exercise intermitted by short impulse breaks (Table 2 and Table 3). Although sometimes difficult to categorize and dependent on the study outcome, the majority of studies (69%) scheduled moderate and moderate-to-high impulse intensity, usually specified by RPE or (less frequently) % maximum tolerable impulse intensity.

Of importance in this context, not all studies that applied superimposed WB-EMS protocols with exercises of moderate to high voluntary intensity installed a corresponding control group that focused on the voluntary exercises. This aspect prevents a reliable assessment of the proper WB-EMS effect in quantitative studies, however.

Unfortunately, 5 projects failed to report adverse effects [34,35,39,42,44] and did not respond to our corresponding queries. Apart from that, no study reported adverse effects; however, this predominately refers to serious adverse effects, while a few studies focus on “abnormal laboratory findings” [18,29,37,38,51,52]. However, these address moderately increased creatine kinase levels post-exercise (no matter how far from thresholds for rhabdomyolysis [57]), a phenomenon that is to be expected after intense resistance-type exercise [58]. Four studies did not report loss to follow-up [34,35,39,44] or respond to our query. Apart from one project that reported a very high lost-follow-up/withdrawal rate due to COVID-19 lockdown [55,56], loss to follow-up averaged between 21% and 0% (Table 2 and Table 3). Finally, attendance rate of the 19 longitudinal projects that reported this outcome was close to 100% due to the opportunity to catch up on missed sessions.

**Table 2 sports-13-00302-t002:** Exercise characteristics of studies that address longitudinal effects.

	Author	Study Length [Weeks]	Superimposed Exercise?	Isolated WB-EMS?	EMS-Sessions n/Week × Length [min]	Exercise/WB-EMS Protocol Impulse Frequency (Hz), -Width (µs), -Duration (s), -Break (s), -Intensity (RPE)	Exercise/Activity in the Control Group(s)	Loss to FU (%)/ Attendance (%)/ Adverse Effects
1	Amaro-Gahete et al. [14]	6	Yes	No	1 × 12–20	WB-EMS: Variable, undulated periodized WB-EMS: 12 and 90 Hz, 350 µs, 4–30 s, 4–30 s, 10–17 [CR 20]	Without EMS	14/96/no
2	Amaro-Gahete et al. [15]	6	Yes	No	1 × 12–20	See above	Standard WB-EMS: 1 × 12–20 min, 85 Hz, 350 µs, 4 s−4 s, RPE 12–17	14/96/no
3	Dörmann et al. [22]	4	Yes	Yes	2 × 20	DRT (see control) superimposed by WB-EMS, 85 Hz, 350 µs, impulse during exercises, RPE ≥ 16 (CR20)	RT: 4 ex., 3 × 8–10 reps RPE ≥ 16 Power: 5 ex., 3 × 5–10 reps/3 × 8 s	21/100/no
4	D‘Ottavio et al. [19]	6	Yes	Yes	2 × 20	10 isometric exercises superimposed by WB-EMS: 350 µs, RPE 14–16: (a) 50 Hz, 4 s−6 s versus (b) 85 Hz, 4 s−4 s	DRT: 7 exercises, 3 × 10 reps 65% 1RM	0/100/no
5	Evangelista et al. [24]	8	Yes	Yes	2 × 20	DRT (see control) superimposed by WB-EMS, 85 Hz, 350 µs, continuous impulse, RPE 7–8 (CR10)	DRT: 2 exercises 3 × 8–12 at 1RM	16/100/no
6	Filipovic et al. [29]	14	Yes	Yes	2 × 9	Squat jumps: 3 × 10 reps superimposed by WB-EMS 80 Hz, 350 µs, 4 s−10 s, up to RPE 18–19 (CR20)	Squat jumps without WB-EMS	0/100/no
7	Filipovic et al. [27,28]	7	Yes	Yes	2 × 9	Squat jumps: 3 × 10 reps superimposed by WB-EMS 80 Hz, 350 µs, 4 s−10 s, RPE 16–19 (CR20)	(1) Squat jumps: 3 × 10 reps without WB-EMS (2) Regular soccer routine only	4/100/no
8	Hussain et al. [30]	8	No	Yes	3 × 20	Swing training (300 swings/week, all groups) + DRT: (12 ex. 2–3 × 2–12 reps,) superimposed by WB-EMS: 85 Hz, 350 µs, 5 s−5 s, 50–80% max. intensity	Swing training only	0/100/no
9	Hussain et al. [31,32]	8	Yes	Yes	3 × 20	Swing training (300 swings/week all groups) + DRT: 12 ex. 2–3 × 2–12 reps, 65–85% 1RM superimposed by WB-EMS, 85 Hz, 350 µs, 5 s−5 s, 50–80% maximum tolerable intensity	(1) Swing training only (2) DRT only	0/100/no
10	Ilbak et al. [33]	12	Yes	Yes	2 × 20	Plyometric exercise (8 ex, 3 × 10–12 reps) superimposed by WB-EMS: 20 Hz, 350 µs, 10 s−10 s, 50–80% maximum tolerable intensity	Plyometric exercise only	0/100/no
11	Jawad et al. [34]	8	No	Yes	3 × 20	Rehabilitation program (19 DRT exercise, 3–4 × 10–20 reps) combined (superimposed?) with WB-EMS (ng)	Rehabilitation program (19 DRT exercise, 3–4 × 10–20 reps	ng
12	Kacoglu et al. [35]	6–4	Yes	Yes	2 × 25	DRT: seated leg press (3 × 20 reps) superimposed by WB-EMS: 100 Hz, 400 µs, 5 s−10 s, RPE 8–9 (CR 10)	DRT: seated leg press (3 × 20 reps)	ng
13	Ludwig et al. [38]	10	Yes	Yes	1 × 20	20 min strength and power training (10 exercises to increase strength, power, strength endurance) superimposed by WB-EMS 85 Hz, 350 µs, 4 s−4 s, RPE 6–7 CR10	20 min strength and power training only	0/97/no
14	Martín-Simón et al. [39]	6	Yes	Yes	1 × 13	3 sessions, 100–140 jumps with 1 session superimposed by WB-EMS: 120 Hz, 350 µs, 5 s−10 s, max. tolerable intensity	3 sessions, 100–140 jumps only	ng
15	Mathes et al. [40]	4	Yes	Yes	3.5 × 60	Cycling at 60% peak power output, superimposed by WB-EMS: 80 Hz, 400 µs, 10 s–2 s, maximum tolerable intensity	Cycling at 60% peak power output only	13/100/no
16	Micke et al. [41]	8	Yes	Yes	2 × ≈25	DRT: 5 ex, 3 × 5–10 reps, RPE > 16 (CR20) superimposed by WB-EMS: 85 Hz, 350 µs, adjusted to exercises 70% max. intensity	DRT only	0/100/no
17	Qin et al. [42]	6	No	Yes	3 × 30	WB-EMS: 85 Hz, 350 µs, 4 s−4 s, RPE 6 CR10 with easy exercises during the impulse phase	DRT: 5 exercises, 3–6 × 5 reps at 80–100% 1RM (?)	20/ng/ng
18	Rappelt et al. [43]	4	Yes	Yes	3 × 20	DRT: Two exercises (squat, glute/ham bridge), 3 × 10 reps, (a) dynamic vs. (b) static, both superimposed by WB-EMS: 85 Hz, 350 µs, 6 s−4 s, RPE 6–8 (CR10)		16/>75/no
19	Sadeghipour et al. [44]	6	No	No	2 × 20	WB-EMS 85 Hz, 350 µs, 6 s−4 s, RPE 14–16 (CR20), with easy exercises during the impulse phase	DRT: 4 ex, 3 × 8–12 reps, 60–80% 1RM Inactive control group	ng
20	Schuhbeck et al. [45]	12	Yes	Yes	1 × 20	6 weeks of static, 6 weeks of dynamic RT exercise supe-imposed by WB-EMS 85 Hz, 350 µs, 4 s−4 s, ≥75% max Intensity (additionally to normal training)	Regular ice-hockey training only	13/100/no
21	Wirtz et al. [51,52]	6	Yes	Yes	2 × 10	Back half squats, 4 × 10 reps to RM, superimposed by WB-EMS, 85 Hz, 350 µs, 5 s−1 s at 70% max. tolerable intensity	Back squats, 4 × 10 reps to RM	0/100/no
22	Wirtz et al. [50]	7	Yes	Yes	2 × 9	Squat jumps: 3 × 10 reps superimposed by WB-EMS 80 Hz, 350 µs, 4 s−10 s, progressive RPE 16–19 (CR20)	(1) Squat jumps: 3 × 10 reps without WB-EMS (2) Regular soccer routine only	0/100/no
23	Zhang et al. [53]	6	Yes	Yes	2 × 20–25	DRT: 4 exercises, 5 × to nRM ^1^ at 85% 1RM superimposed by WB-EMS 85 Hz, 350 µs, EMS during sets, 60–100% device capacity	DRT without WB-EMS	17/100/no
24	Zink-Rückel et al. [55,56]	16	No	Yes	1 × 20	WB-EMS 85 Hz, 350 µs, 6 s−4 s, RPE 6–7 CR10 with easy golf specific exercises during the impulse phase	Regular golf routine only	33 ^2^/97/no

1RM: one repetition maximum; DRT: dynamic resistance exercise training, Ex: exercises; FU: follow-up; nRM: non-repetition maximum [59]; Reps: repetitions; RPE: rate of perceived exertion; WB: whole body (most or all major muscle groups methodologic quality. ^1^ Set endpoint: velocity as assessed during the set dropped by 10%. ^2^ Due to COVID-19 lock-down, 8 participants quit the study, and one participant was unable to attend the 16-week follow-up assessment.

**Table 3 sports-13-00302-t003:** Exercise characteristics of studies that address acute effects.

	Author	Conditions [n]	Superimposed?	WB-EMS Protocol Impulse Frequency (Hz), -Width (µs), -Length (s), -Break (s), -Intensity (RPE)	Exercise/Activity in the Control Group	Loss to FU (%)/Adverse Effects
1	Berger et al. [16]	4	no	WB-EM: 85 Hz, 350 µs, 4 s−4 s, 4 trials each up to maximum tolerable intensity	-----------	0/none
2	Buonsenso et al. [18]	1	yes	DRT: 3 exercises, 4 × 12 reps (12RM) 20 min, superimposed by WB-EMS 85 Hz, 350 µs, 4 s−4 s, RPE 6–7 (CR10)	Resistance exercise	0/none
3	Boccia et al. [17]	2	yes	Isom. RT: 5 exercises, 6 × 30 s (6 s contraction 4 s rest) superimposed by WB-EMS, 85 Hz, 350 µs, 6 s−4 s, maximum tolerable intensity	5 isometric exercises (not all similar to WB-EMS)	0/none
4	De Arrilucea et al. [20]	2	yes	FIFA11+ warm up protocol: 25 min superimposed by WB-EMS 20 Hz, 350 µs, 6 s−4 s, 60–100% device capacity	FIFA11+ warm-up only	0/none
5	De La Camara et al. [21]	3	no	WB-EMS: 20 min, 1 Hz, 350 µs, continuous stimulation, most comfortable intensity	(1) Passive recovery, (2) cycling	ng/none
6	Dote-Montero et al. [23]	4	yes	WB-EMS: 100 Hz, 200–400 µs, 5 sets of 6 s−30 s (duty cycle 99%), maximum tolerable intensity	2 × 3 rep one leg squat	0/none
7	Fernández Elías et al. [26]	3	yes	DRT: Squat-, bench press, 5 × 5 reps, 90% 1RM superimposed by WB-EMS, 85 Hz, 250/350 µs, continuous WB-EMS or 85 Hz, 250/350 µs, 1 s (concentric)−2 s (eccentric)	DRT without WB-EMS	0/none
8	Fernández-Elías et al. [25]	2	yes	FIFA11+ warm up protocol: 25 min, 20 Hz, 350 µs, 6 s−4 s, 60–100% device capacity	FIFA11 + warm up only	0/none
9	Kemmler et al. [36]	2	no	WB-EMS: 16 min, 85 Hz, 350 µs, 4 s−4 s, RPE 15 (CR20) with easy dynamic exercises (2 × 8 reps without additional load)	The same 5 easy dynamic exercises only	0/none
10	Kemmler et al. [37]	5	Yes no	Endurance-type protocols: Cross-Trainer (30 min at 75% VO_2_max) superimpo-sed by continuous WB-EMS stimulus 350 µs at 7 Hz or 85 Hz, RPE 7 at CR 10 DRT-type protocols: WB-EMS (20 min 85 Hz, 350 µs, 4 s−4 s, RPE 7 at CR10) with easy dynamic exercises (2 × 8 reps without additional load)	30 min cross trainer only The same easy dynamic exercises only	0/none
11	Teschler et al. [46]	1	no	WB-EMS: 85 Hz, 350 µs, 6 s−4 s, moderate-high intensity	No exercise	0/none
12	Wahl et al. [49]	3	yes	Cycling step test to volitional exhaustion (27 min) superimposed by WB-EMS, 30 Hz or 85 Hz, 400 µs, 10 s−5 s, maximum tolerable intensity	Cycling step test (29 min) without WB-EMS	0/none
13	Wahl et al. [47,48]	3	yes	Cycling (60 min at 70% peak power output) superimposed by WB-EMS, 60 Hz, 400 µs, continuous stimulation, maximum tolerable intensity	(1) Cycling only (2) EMS only	0/none
14	Zink-Rückel et al. [54]	2	no	Seven easy golf swing exercises (1 × 10 reps) with WB-EMS (85 Hz, 350 µs, 4 s−4 s, RPE 3–4 at CR10)	Seven easy golf swing exercises (1 × 10 reps)	9/none

### 3.4. Methodologic Study Quality

Table 1 displays the methodologic quality of the studies included according to PEDro [11]. In summary, the PEDro score of longitudinal intervention studies (Table 2) ranged from 3 to 7 score points out of a maximum of 10 score points. While one study applied a single-group protocol and was not rated [16], the PEDro score for studies that focus on acute WB-EMS effects (Table 3) ranged from 4 to 8 score points. The main reasons for low ratings are attributable to the criteria of “allocation concealment”, “blinding of participants”, and/or “blinding of therapists”. However, bearing in mind that the latter two criteria are not reliably implementable in exercise (including WB-EMS) studies, 8 score points should be considered as a realistic maximum for this topic.

### 3.5. Evidence Map Data

Figure 3 and Figure 4 provide a quick overview of outcomes addressed by acute and longitudinal WB-EMS application. In summary, 39 studies addressed 79 outcome categories (e.g., isometric strength) and more than 300 single outcomes (e.g., maximum isometric strength of leg flexors). While most studies focused on one or two outcome categories, three projects with two publications each [27,28,51,52,55,56] covered four outcome categories. Thirty-one studies focused on performance-related outcomes (79%; [14,15,18,19,20,22,23,24,25,26,27,28,29,30,31,32,33,34,35,38,39,40,41,42,43,44,45,49,50,51,52,53,54,55,56]), four studies (11%; [21,43,51,52,56]) addressed regeneration-related outcomes, and eight studies (22%; [14,15,24,27,28,35,42,44,55]) reported outcomes related to anthropometry. However, anthropometry was considered as a primary or core study endpoint [24,44,55] in only three studies. In parallel, only one [16] of the 14 studies (36%; [16,17,18,26,27,28,29,34,37,40,47,48,49,51,56]) that reported health- and safety-related outcomes specifically focused on this outcome. Lastly, 16 studies (41%) provided explanatory outcomes [14,15,17,25,26,27,29,36,37,43,46,47,48,50,51,52,55], predominately to provide deeper insight into changes in performance parameters also reported (e.g., changes in running economy to explain changes in time to failure). However, three corresponding studies [17,36,46] focused on WB-EMS-induced energy expenditure/oxygen consumption that was considered as an explanatory outcome for fat tissue reduction.

**Figure 3 sports-13-00302-f003:**
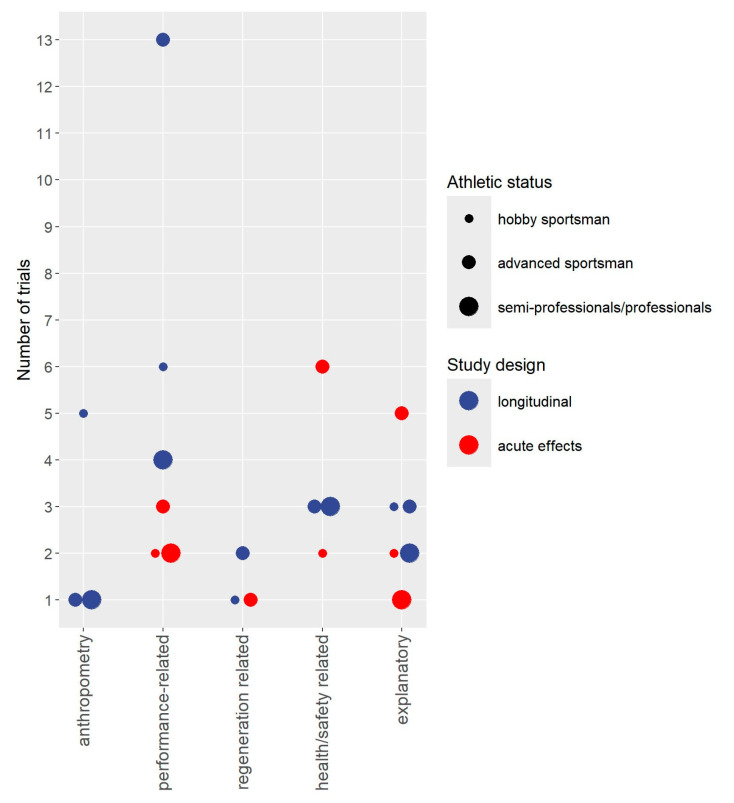
Outcomes addressed by WB-EMS study with sportspeople and athletes.

With respect to the 31 studies that reported performance-related outcomes, 25 addressed strength (80%; [17,18,19,22,23,24,26,27,28,29,30,31,32,34,35,38,39,40,41,42,43,44,45,49,51,52,53,56]), 10 power (32% [19,22,28,34,40,41,43,49,51,52,53]), 18 jumping (58%; [14,15,17,20,22,25,26,29,33,35,39,40,41,43,45,49,51,52,53]), ten sprinting (32% [20,22,25,29,33,35,40,41,45,51,52]), six agility (19%; [22,26,29,33,41,51,52], six endurance (19% [14,15,27,28,40,49,50]), five anaerobic power (16% [20,25,33,35,49]), and only one each flexibility or balance (3% [18] [33]) (Figure 4). Further five studies (16%) [20,29,30,45,54] reported sport-specific performance outcomes (e.g., kicking or shot velocity, clubhead speed).

**Figure 4 sports-13-00302-f004:**
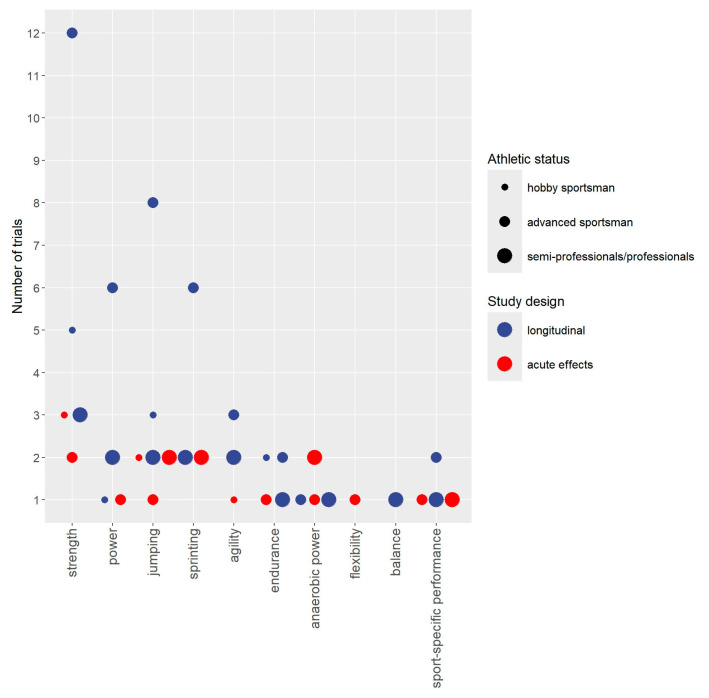
Performance reported outcomes addressed by WB-EMS study with sportspeople and athletes.

## 4. Discussion

Less surprisingly, the present study is the first evidence map that aimed to provide an overview of outcomes addressed by WB-EMS application in sportspeople and athletes. The study completes a previous evidence map on cohorts and outcomes addressed by longitudinal WB-EMS application but focused on “non-athletic” cohorts. We considered a corresponding separation and dedicated processing of the two topics to be important, as training objectives and correspondingly addressed study outcomes obviously vary between the groups. This particularly refers to outcomes related to performance or regeneration that are increasingly important regardless of the performance level of the exerciser, while its relevance for non-sportive cohorts is limited [60]. In summary, we identified 36 studies disseminated in 39 publications that focus on 78 outcome categories (Figure 4) with more than 300 single, albeit partially overlapping, outcomes. As expected, most studies focused on performance-related outcomes; a few addressed regeneration, ultimately a domain closely related to athletic performance. Apart from a few studies that particularly focus on this issue (e.g. [24,55]) anthropometry and safety/adverse effects were merely reported as a subordinate outcomes. Only one (longitudinal) study [56] determined dedicated orthopedic diseases frequently associated with the given type of exercise as an experimental outcome. This lack of studies in the area of injury prevention and prophylaxis of adverse conditions should be considered an important evidence gap. This conclusion is based on the fact that as a time-efficient resistance exercise method to address upcoming or present sport-specific orthopedic limitations [61,62,63], WB-EMS might be an even more striking argument for application, particularly in endurance-type exercise, than performance increases alone. While evidence gaps related to performance parameters will be discussed in more detail below, another less addressed topic of nevertheless crucial importance for advanced sportspeople and athletes is the acute effect of WB-EMS on regeneration. So far, only de la Cámara et al. [21] has looked at this issue, comparing 20 min of (very) low frequency (1 Hz) continuous WB-EMS with “most comfortable stimulus intensity” with 20 min of passive regeneration (laying in a supine position) and 20 min of cycling (60 rpm) at 20 W immediately post-exercise (5 min of intense rowing). Anthropometry or body composition was also rarely determined by studies with sportspeople. Although there might be some area of application, e.g., longitudinal effects on body shaping in recreational sportspeople, we feel that the multitude of studies with “non-athletic cohorts” [64] might be widely transferable to the more sportive cohorts addressed in this article. Nevertheless, some specific aspects, e.g., acute effects on cutting weight in types of exercise with weight classes, might be of interest. In summary, however, we would not place the focus of WB-EMS research with sportspeople and athletes in this area.

Reviewing performance-related outcomes in more detail, the vast majority of studies addressed strength, power, jumping (which can be largely subsumed under “power”), sprinting, and agility (predominately determined by multidirectional sprint tests), types of exercise that should be sensitive to the resistance-exercise character of standard WB-EMS protocols [4] predominately applied by the studies (Table 2 and Table 3). To a lesser extent this may also refer to acute or longitudinal changes in anaerobic power (as determined by repeated sprint ability or the Wingate test), a performance-related outcome addressed by four studies [20,25,33,35] presumably as a subordinate study endpoint. Only three (longitudinal) projects with four studies [14,15,27,50] focused on endurance-related outcomes (e.g., time to failure, rel. VO_2_max/peak) in recreational runners [14,15] or advanced and professional soccer players [27,50]. While Filipovic et al. [27] and Wirtz et al. [50] scheduled the similar brief jumping exercise protocol (3 × 10 reps) superimposed by WB-EMS, Amaro-Gahete et al. [14,15] applied an undulated periodized WB-EMS program with a constant duty cycle of 50% with varying impulse phases (4 s−30 s) (Table 2). Probably as an experimental outcome, Buonsenso et al. [18] addressed (among others) flexibility after a single bout of DRT superimposed by standard WB-EMS in sport students, and Ilbak et al. [33] determined changes in balance after 12 weeks of plyometric exercise (8 ex, 3 × 10–12 reps) superimposed by WB-EMS in young basketball players.

In summary and as to be expected, the condensed outcome summary listed above revealed a large number of evidence gaps, the significance of which varies depending on the individual discipline-specific perspective. Nevertheless, some general evidence gaps should be addressed in more depth. (1) Of importance, only one study [53] focused on participants with a background of resistance training exercise (Figure 2). Bearing in mind that WB-EMS can be considered as a resistance-type exercise in its standard application [4], this finding is surprising at first but might indicate that many researchers regard the additive effects of WB-EMS on strength development as rather limited in specifically pretrained cohorts. However, there is evidence from local EMS that neuromuscular electrical stimulation (NMES) of both quadriceps femoris muscles is an effective supplement to weight training even in elite weightlifters [65]. (2) In parallel, only a handful of longitudinal projects addressed participants from endurance [14,15] or precision sports [55,56]. While Amaro-Gahete et al. focused on outcomes closely related to endurance performance in his cohort of hobby runners, Zink-Rückel et al. [55,56] failed to assess the prespecified golf-specific outcomes (i.e., average golf score for five rounds on an 18-hole course) due to COVID-19 lockdown in Bavaria. (3) As already stated, there is a conspicuous absence of longitudinal studies on health-related outcomes. This also includes parameters considered widely irrelevant for the given sport-specific performance but crucial for preventing breakdown due to pain complaints or injuries. (4) In parallel, only a few studies evaluated the effect of WB-EMS on outcomes related to acute regeneration. (5) Further, only one study focused on WB-EMS as a warm-up. (6) Another evidence gap refers to flexibility and balance as study outcomes. Although we (speculatively) do not expect significant positive effects of WB-EMS on flexibility and/or balance, it should at least be possible to rule out negative effects.

We acknowledge that some of the features and limitations of this evidence map may be challenging for interpreting the results clearly. (1) In contrast to our last evidence map on “non-athletic cohorts”, the present study focuses on people with a history of regular exercise. Reviewing the literature, however, we did not find a reliable definition of such a cohort. While the minimal criteria of “athletes” suggested by Araujo et al. [9] are inadequate for our approach (e.g., “to be formally registered in a local, regional or national sport federation” or “to have sport training and competition as his/her major activity (way of living) or focus of personal interest”), we finally decided to apply an eligibility criterion of ≥2 sessions/week during the last 2 years without athletic competitions. Since some authors do not clearly report weekly frequency and history of exercise participation, in cases of doubt, authors were contacted and requested to state whether their publication was suitable for inclusion. Unfortunately, three authors could not be reached or were unable to respond; thus, we cannot be sure that only or all eligible articles were included. This particularly refers to “hobby sportspeople”, while advanced sportspeople (including sport students) and semi-professionals/professionals were identified more reliably. (2) Although to a lower degree compared to non-athletic cohorts [3,64], the included cohorts of sportspeople cannot be considered as a homogeneous cohort. In detail, the study participants range from recreational/hobby runners [14,15] to professional soccer players [29], with corresponding consequences for differences in lifestyle. Of note, the number of projects that focused on elite “athletes” was quite low, and all such projects addressed team sports. In contrast, the best-represented cohort are physical education/sports students, categorized as “allrounders” and advanced sportspeople. However, this cohort might be more heterogeneous because, apart from common exercises scheduled in the degree programs, many students have different competitive sports of different discipline backgrounds. (3) In this context, however, we have to admit that in a few cases our categorization of outcomes might also be considered inadequate. For example, after intense discussion among the reviewers, the trial of Berger et al. [16] on adjustment effects of stimulus intensity after multiple consecutive EMS sessions was subsumed under “safety/health-related outcomes”, while this aspect might also address “performance”. (4) Only a few studies provided a reliable hierarchy of outcomes. Accordingly, it is not always clear whether the study really focused on the given outcome or regarded it as an experimental, less important outcome. This limitation might be even more relevant when study effects are addressed. (5) Methodological quality was rated by the PEDro scale that is not perfectly applicable for randomized cross-over trials and, in particular, non-randomized trials (Table 1).

## 5. Conclusions

In summary, the present evidence map provides a robust and comprehensive overview of topics and outcomes addressed by WB-EMS in regularly exercising people, competitive sportspeople, and semiprofessional/professional athletes. As might be expected with such a novel exercise technology, there are far more research gaps than issues that have already been addressed. Although the relative relevance of the research gap might predominantly depend on the individual priorities and research interests, we suggest that some WB-EMS research-specific issues should be given more attention in the near future. This includes injury prevention, particularly in elite team sports, as well as in endurance athletes who are prone to orthopedic complaints (e.g., low back pain). It also includes muscle regeneration after competitive exercise and/or training periods with intense exercise, as well as research issues related to the most effective exercise protocol for achieving different sport-specific goals.

## Data Availability

The datasets generated and/or analyzed during the current study are available from the corresponding author on reasonable request.

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
