# Peer review of "Outcomes Addressed by Whole-Body Electromyostimulation Trials in Sportspeople and Athletes—An Evidence Map Summarizing and Categorizing Current Findings"

_sports, 2025, doi:10.3390/sports13090302_

Round 1

Reviewer 1 Report

Comments and Suggestions for Authors

The article provides a comprehensive and systematic review of outcomes addressed by whole-body electrostimulation trials in sportspeople and athletes. The evidence map is well-structured, adhering to PRISMA guidelines, and offers valuable insights into the current state of research, including gaps and future directions. The methodology is robust, and the results are clearly presented. However, there are areas where clarity, detail, and consistency could be improved to enhance the article’s overall quality and impact.

  • Pg 1; Ln 22: You mentioned “78 outcome categories with more than 300 single outcomes”, but it is unclear how these categories were defined or grouped. A brief explanation may improve clarity.
  • Pg 1; Ln 3-31: The statement about evidence gaps in the acute effects of WB-EMS on regeneration is vague. Specify which aspects of regeneration are understudied (e.g., muscle recovery, inflammation markers).
  • Pg 2: The claim about the WB-EMS being “joint friendly” is not strongly supported by the reviewed studies. Provide specific evidence or qualify the statement as a hypothesis.
  • Pg2: Also, being time-efficient, it lacks quantitative comparisons to traditional interventions. Cite studies or metrics (session duration vs outcomes) to substantiate this.
  • Pg 2: The justification for separating “non-atheletic” and athletic cohorts is weak. Elaborate on why these groups require distinct evidence maps, citing specific differences in training objectives or physiological responses.
  • Pg 2; Ln 69: The eligibility criteria for “advanced sportspeople” are not clearly defined. Provide explicit thresholds (e.g., training volume, competition level) to avoid ambiguity.
  • Pg 3; Ln 130: The RoB assessment using PEDro is mentioned, but the results of this assessment are not summarized in the results section. Include a table or summary of bias ratings for transparency.
  • Pg 12: The statement about the adverse effects is contradictory. It claims “no study reported adverse effects,” but later mentions “abnormal laboratory findings. Please clarify whether these findings are considered adverse effects.
  • The conclusion oversimplifies the evidence gap. Specify priority area for future research (e.g., Elite athletes, injury prevention) rather than general statements
  • The summary paragraph in the discussion should be moved under the “conclusion” subheading.
  • Page numbering don’t follow the true order. Revise

Reviewer 2 Report

Comments and Suggestions for Authors

Please consider the following as an attempt to assist in the refinement of a manuscript that provides a very solid foundation from which to build. 

Line 3: Consider clarifying “evidence map” for unfamiliar readers by adding a short phrase such as: “– An evidence map summarizing and categorizing current findings.”

Lines 16–18: “highly customizable training technology that particularly attracts sportspeople...” – This could be simplified for clarity. Suggested: “a customizable training method popular among athletes seeking performance enhancement, faster recovery, and injury prevention.”

Lines 21–22: “78 outcome categories with more than 300 single outcomes” – It would be helpful to briefly explain what defines an "outcome category" vs. a "single outcome".

Lines 38–40: Consider citing a broader range of sources beyond \[1, 2] to support the claim about WB-EMS being “time-effective, joint-friendly”.

Line 43: “Summarizing the present research in this area is problematic…” – Consider specifying what makes summarization difficult: Is it heterogeneity in study design, population, outcome measures?

Lines 47–53: Consider briefly explaining how evidence mapping differs methodologically from meta-analysis in terms of inclusion criteria or synthesis goals.

Lines 57–58: “...as an adjuvant to their exercise training” – consider replacing “adjuvant” with “supplementary” for clarity unless targeting a primarily clinical audience.

Lines 65–66: Good to see PROSPERO registration. Consider stating the registration date in addition to the ID.

Lines 70–71: “...training frequency of two and more sessions per week” – change to “two or more sessions” for clarity and grammatical correctness.

Line 78: Clarify whether exclusion of local EMS studies was due to mechanistic differences or scope limitation.

Lines 85–86: Consider explaining why bachelor/master theses were excluded but dissertations included.

Lines 96–100: The search string could be moved to a supplementary section for readability. Currently dense and better suited to appendices.

Lines 129–134: You mention PEDro scoring but should acknowledge its limitations for crossover or non-randomized studies, as you do later in the discussion.

Lines 150–158: Consider summarizing the demographics more visually here—e.g., “Participants in the reviewed studies ranged from 15–43 years old; most were male, and the majority were advanced sportspeople or hobby athletes.”

Line 153: "and for the analysis" – remove or clarify this phrase.

Line 176: “Autor” appears to be a typo or a leftover from a data table. Correct to “Author”.

Lines 179–189: The participant classification is useful, but the definitions could be clarified earlier in the Methods for consistency.

Lines 200–219: Dense methodological detail on WB-EMS protocols would be better understood in table form with a clearer legend (Tables 2 and 3 are referenced but not included in this snippet).

Lines 269–278: This section successfully contextualizes the review. Consider bringing up implications for practice sooner.

Lines 285–289: The point about injury prevention is strong but could be better linked to training demands and chronic load management.

Line 302: “some specific aspects ,e.g.” – remove the space before the comma.

Lines 327–331: The lack of resistance-trained cohorts is an important observation—consider expanding briefly on why this group may benefit or not from WB-EMS.

Lines 340–342: “Although we do not expect significant positive effects…” – clarify if this is based on prior research or authors’ opinion.

Lines 344–383: These limitations are well acknowledged. However, consider tightening to avoid redundancy (e.g., lines 374–380 restate earlier points).

Line 366: “study participants range from recreational runners \[13,14]...” – consider rewording for parallel construction.

Conclusion (Lines 384–385)

This section appears unfinished. Add a concise summary of the main findings and their implications for future research and clinical/practical application. For example:

“This evidence map reveals a substantial concentration of WB-EMS research on strength and power in moderately active populations. Future studies should explore its utility in endurance and resistance-trained athletes and evaluate its efficacy for injury prevention and recovery enhancement.”
